# Spatial and temporal Trend Analysis of Long Term rainfall records in data-poor catchments with missing data, a case study of Lower Shire floodplain in Malawi for the Period 1953-2010

Rabee Rustum<sup>1</sup>, Adebayo J. Adeloye<sup>2</sup>, Faidess Mwale<sup>3</sup>
<sup>1</sup>School of Energy, Geoscience, Infrastructure and Society, Heriot-Watt University, UK, Dubai Campus, Dubai International Academic City, P O Box 294345, Dubai, UAE. <u>r.rustum@hw.ac.uk</u>
<sup>2</sup>School of Energy, Geoscience, Infrastructure and Society, Heriot-Watt University, UK, Edinburgh Campus, EH14 4AS <u>a.j.adeloye@hw.ac.uk</u>
<sup>3</sup>Ditle Device the Machine Machine Content of Content

<sup>3</sup>Faidess Dumbizgani Mwale, Head of Department & Senior Lecturer in Hydrology and Water Resources Management Department of Civil Engineering, University of Malawi - The Polytechnic, P/Bag 303, Blantyre 3, Malawi. <u>fmwale@poly.ac.mw</u>

Correspondence to: Rabee Rustum (r.rustum@hw.ac.uk)

- Abstract. This paper investigated the long-term trends in precipitation from 16 stations located in the lower Shire catchment in Malawi over the period 1953-2010. Annual trend analysis was first considered, and in order to take into account seasonality and serial correlation, the different months of the year are considered. Trend significance was determined using the nonparametric Mann-Kendall (MK) test statistic while the determination of the trends magnitudes was achieved using Sen's slope method. The homogeneity of trends was examined using the Van Belle and Hughes method. The results indicate that
- annual precipitation has increased, whereas, monthly precipitation revealed an upward trend in wet seasons (November to April) and a downward trend in dry seasons (May to October). The monthly peak trend analysis has shown upward trend in rainy months at all stations.

Keywords: Climate change; Watershed hydrology; statistical trend analysis; Mann–Kendall test; Sen's slope estimator; Van
 Belle and Hughes.

# **1** Introduction

The impact of climate change on water resources is felt worldwide but its effects are more overwhelming in places where flooding or drought takes place (Howard et al. 2016; Arnell et al. 2016; IPCC 2014; Bhave et al. 2016). However, the foremost influences are witnessed on the water cycle and its effect on domestic use, flood control, irrigation, etc (Daccache et al. 2017;

Simonovic 2017; Makwiza et al. 2015). Changes in rainfall patterns might affect future planning in terms of housing and other urban facilities, proposed irrigation projects, land use, insurance and other activities that assume the climate will not change over project life. In addition, intensification in rainfall may lead to increased frequency of floods, landslides, soil loss, sediment

transport, and would have consequences for aquifer recharge and the general water quality situation (Khan et al. 2016; Pina, Amaury, and Francois 2016; Hattermann et al. 2017; Scherler et al. 2016).

Therefore, to protect water resources availability and assess future land use, efforts have been devoted to study climate change, especially as it affects rainfall. These studies aim to understand the trend of change in rainfall patterns by analysing long term historical rainfall and runoff data, which then forms the basis of forecasting future scenarios.

For example, Jana et al. (2017) analysed spatio-temporal rainfall change scenarios in the 20<sup>th</sup> century (1901–2000) over Bundelkhand, India. They found that decreasing rainfall trends during monsoon seasons and increasing trends during pre and post-monsoon seasons are an indication of shifting rainfall patterns away from a typical seasons.

Furthermore, Engström and Waylen (2017) analysed historical (1952–2014) precipitation and streamflow data from 18 basins in southeast United States for long-term changes in the hydroclimatology. They found that there is a change in the precipitation-runoff relationship in the late 1990s where there is a decreased precipitation storage and consequently increased streamflow.

15

25

10

5

Buendia et al. (2016) analysed the hydro-climatic trends at the annual and monthly scales in three nested sub-catchments in a central Pyrenean basin, Spain, using the Non-parametric Mann–Kendall statistic for data from 1965 to 2009. The results demonstrate that upward trends were detected for temperature and potential evapotranspiration, particularly during summer months (June to August) and winter months (December to February). Precipitation trends indicated a decrease, particularly for

20 February and July. Results also indicated that a change in annual runoff took place in the 1980s.

Paul et al. (2017) used both parametric and non-parametric approaches to identify the trends at different temporal scales of the Rajahmundry city rainfall, lower Godavari basin, India, during the period 1960–2013. They witnessed a negative trend at a weekly scale during the monsoon months. For example, the magnitude of Sen's slope was observed to be negative during the months of April–September.

Therefore, trend analysis can be used to examine if there is any significant change in climatological parameters and several statistical techniques now exist to detect trends in hydro-meteorological data. These techniques can be classified into parametric and non-parametric approaches. Parametric methods make assumptions about the form of probability distribution

30 of the variable, i.e. principally that the variable has a Gaussian distribution, which often invalidates their applications to non-Gaussian distributed variables. Additionally, even where it can be established that the variable does follow the normal distribution, the estimation of the parameters of the distribution can be hampered, i.e. subject to large biases and variability, due to the typically short data records that are available.

This work aims to find the trends in precipitation in the Shire Basin, Malawi, not only in the annual total precipitation but also in the total monthly precipitation, in addition to the trends of monthly peaks. The study utilised a record of daily data that spanned 58 years (1953-2010). The daily records from which the monthly and annual records were generated contain numerous

5 gaps and missing values which require a scientific infilling before performing any trend analysis. This infilling exercise has been performed and reported elsewhere (Mwale, Adeloye, and Rustum 2012); hence only a brief mention of this will be made here.

The procedures implemented in this paper are based on the non-parametric Mann-Kendall test to detect a monotonic trend of a time series. The magnitude of the trend, and the slope of the linear trend, were estimated with the nonparametric Sen's method, which is not significantly affected by single data errors or outliers (Thenmozhi and Kottiswaran 2016; Bouza-Deaño, Ternero-Rodríguez, and Fernández-Espinosa 2008). The spatial and temporal homogeneity of trends were examined using Van Belle and Hughes (Kahya and Kalayci 2004). The following sections discuss these methods in detail.

#### 2 Statistical Trend analysis techniques

# 15 2.1 Mann-Kendall Test

The Mann–Kendall (MK), commonly known as the Kendall's tau statistic, is a non-parametric test used for trend analysis. Mann (1945) first used this test and Kendall (1975) derived the test statistic distribution. The test has been suggested by the World Meteorological Organisation (WMO) to assess trends in environmental data time series (WMO 2014) as the test is suitable for cases where the trend may be assumed monotonic and therefore no seasonal aspects are presented in the data. The

- method is simple, does not require assuming normality, robust against outliers and can handle missing values (Hess, Iyer, and Malm 2001). Accordingly, when compared to parametric test like t-test, the Mann-Kendall test has a higher power for non-normally distributed data, which are normally presented in hydrological data (Yue and Pilon 2004). It is worth to mention that some researchers (e.g. Sang et al., 2014) highlighted the need for pre-whitening of the time series data before conducting any trend analysis test, however, if the sample size is more than 70 then serial correlation does not affect the MK test (Basistha, 2014)
- Arya, and Goel 2009)

The Mann–Kendall test calculates the slope of the line formed by plotting the variable of interest against time, but only considers the sign and not the magnitude of this slope. Hence, the MK test statistic is calculated from the sum of the signs of the slopes. The statistic S is:

$$S = \sum_{i=1}^{n-1} \sum_{j=i+1}^{n} \operatorname{sgn}(x_j - x_i)$$
(1)

15

Where, n is the number of data point,  $x_j$  is the j<sup>th</sup> observation and  $x_i$  is the i<sup>th</sup> observation where j > i. The sgn(.) can be estimated as:

$$\operatorname{sgn}(x_{j} - x_{i}) = \begin{cases} 1 & if \quad (x_{j} - x_{i}) > 0 \\ 0 & if \quad (x_{j} - x_{i}) = 0 \\ -1 & if \quad (x_{j} - x_{i}) < 0 \end{cases}$$
(2)

The Mann-Kendall test is based on the null hypothesis that a sample data is independent and identically distributed, which
means there is no trend in the data points. Thus, if the null hypothesis H<sub>o</sub> is accepted at the significant level α, then the mean and variance of the S statistics are given by Kendall (1975) as it is approximately normally distributed, mean (S) is zero.

$$mean(S) = 0 \tag{3}$$

In the case where there are no ties in either ranking, the distribution of S may be well approximated by a normal distribution 10 with mean zero and variance as stated in Equation 4.

$$Var(S) = \frac{n(n-1)(2n+5)}{18}$$
(4)

And in the case of ties, the variance of S is more complicated as in Equation 5.

$$Var(S) = \left\{ \frac{n(n-1)(2n+5)}{18} - \sum t_i(t_i-1)(2t_i+5) - \sum u_i(u_i-1)(2u_i+5) \right\} + \frac{1}{9n(n-1)(n-2)} \left\{ \sum t_i(t_i-1)(t_i-2) \right\} \left\{ \sum u_i(u_i-1)(u_i-2) \right\} + \frac{1}{2n(n-1)} \left\{ \sum t_i(t_i-1) \right\} \left\{ \sum u_i(u_i-1) \right\}$$
(5)

Where  $t_i \mbox{ is the number of data point for } i^{\mbox{th tie.}}$ 

The normal approximation, Z statistics, is stated in Equation 6.

$$Z = \begin{cases} \frac{S-1}{\sqrt{Var(S)}} & \text{if } S > 0\\ 0 & \text{if } S = 0\\ \frac{S+1}{\sqrt{Var(S)}} & \text{if } S 

10

A positive value of S indicates that there is an increasing trend and vice versa. However, the absolute value of Z is compared with the standard normal cumulative distribution to detect if there is any trend at the selected level of significance ( $\alpha$ ). The trend is said to be decreasing if Z is negative and increasing if Z is positive. H<sub>0</sub>, the null hypothesis of no trend, is rejected if the shealest evelop of Z is compared with a standard normal cumulative distribution to detect if there is any trend at the selected level of significance ( $\alpha$ ). The trend is said to be decreasing if Z is negative and increasing if Z is positive. H<sub>0</sub>, the null hypothesis of no trend, is rejected if

5 the absolute value of Z is greater than  $Z_{1-\alpha/2}$ , where  $Z_{1-\alpha/2}$  is obtained from the standard normal cumulative distribution tables.

## 2.2 Sen's slope estimator

As stated previously, the Mann–Kendall test only indicates the direction but not the magnitude of significant trends. Thus, the magnitude is usually determined by Sen's test (Sen 1968) which is also a nonparametric technique. The method uses a linear model to calculate the change of slope and the variance of the residuals should be constant in time. The function of Sen's model is:

$$f(t) = Qt + B \tag{7}$$

Where Q is the slope and B is constant

To calculate the model parameters, B and Q, the slope between any two observations is calculated as in Equation 8.

$$Q_{i} = \frac{x_{j} - x_{k}}{j - k}; \text{ where } j > k$$
(8)

15 Therefore, the total number of slopes, N, is estimated as:

$$N = \frac{n(n-1)}{2}$$

The slope of Sen's test Q is the median of these Qi slopes.

$$20 \quad Q = median[Q_i]_{i=1}^N \tag{9}$$

The intersect B is the median of n values of difference  $x_i$ -Qt<sub>i</sub>

$$B = median \left[ x_i - Qt_i \right]_{i=1}^n \tag{10}$$

5

20

#### 2.3 Test of Homogeneity of trends

Trend results should be consistent between stations, seasons and stations-seasons. For example, by applying the MK test, each month or each station is dealt with separately, thus the results of trend analysis may not be significant, but when the catchment is considered as a whole, the trend might be highly significant. Therefore, one figure should represent all stations at a particular month, and another figure to represent the trend of all months in a particular station, and a third figure to represent the catchment wise. These figures can be generated by analysing the trend statistics, Z statistics in case of MK test, to get a single trend

figure. The most widely used homogeneity test of trend is the Van Belle and Hughes (van Belle and Hughes 1984). This test has been widely applied in Hydrology (Dabanlı et al. 2016; Akinsanola and Ogunjobi 2017; Kahya and Kalayci 2004).

10 The van Belle and Hughes (1984) procedures are based on dividing the sum of the squares using the Chi-square tests to get the trend homogeneity between months, between stations and station-month interactions.

$$\chi^{2}_{\text{hom}\,ogeneous} = \chi^{2}_{total} - \chi^{2}_{trend} \Rightarrow \chi^{2}_{\text{hom}\,ogeneous} = \sum_{i=1}^{m} (Z_{i})^{2} - m(\check{Z})^{2}$$
(11)

$$\check{Z} = \frac{1}{m} \sum_{i=1}^{m} Z_i$$
(12)

(m=12 for homogeneity between months, m= number of stations for homogeneity between stations, and m=m\* number of stations for station-month interaction).

According to Kahya and Kalayci (2004), if the Chi-square (homogenous) exceeds the predefined  $\alpha$  level critical value for the Chi-square distribution with (m-1) degrees of freedom, the null hypothesis of homogeneous trend must be rejected and in this case, different seasons or stations show dissimilar trends, otherwise, the Chi-square (trend) is referred to the chi-square distribution with 1 degree of freedom to test a common trend.

However, this is a complex procedure and may not be understood by ordinary readers; instead, homogeneity is tested graphically by plotting the Z values over all stations and all months. This provides an overview if the stations or the months are homogeneous, or heterogeneous.

# 25 3 Materials and Methods

#### 3.1 Study Area

The Shire Basin is located in the southern region of Malawi (Figure 1). The Lower Shire floodplain is the most vulnerable area to flooding and thus the most affected in Malawi. Its biophysical vulnerability arises from heavy rainfall in the upper and

5

middle catchments of the Shire River and from the Ruo catchment on its main tributary. Although the floodplain rainfall is only around 600 mm annually, the upper and middle Shire River catchments receive annual rainfall of around 900 mm and the Ruo catchment has average annual rainfall in excess of 2,000 mm. The population is predominantly rural and has the highest levels of poverty in the country. Livelihoods are intricately linked to natural resources, especially water, which is why most of the settlements are along river courses. Due to resource constraints, the catchment has very limited investment in structural measures for flood control (Mwale, Adeloye, and Rustum 2014).

## 3.2 Data collection and handling

Daily rainfall Data from 16 rain gauges, collected by the Department of Climate Change and Meteorological Services
(DCCMS), were used in this study. The DCCMS collects rainfall data using manual rainfall gauges. The historical periods of data provided for this study are presented in Table 1, from which it is clear that the periods are non-uniform. Several of the records are missing and, although this is not evident in Table 1, some of the missing periods do overlap thus, making the use of traditional infilling approaches such as regression impossible. Therefore, the self-Organizing map (SOM) was first used to infill the missing data in order to generate a complete record. This task has been published in (Mwale, Adeloye, and Rustum)

- 2012). Therefore, 16 stations with sufficient length of records during the period of Jan. 1953– Dec.2010 were selected for the present analysis. The 58 year record for the 16 stations meets minimum conditions of 25 years set by (Burn and Hag Elnur 2002) that ensures validity of the trend results statistically. Annual data statistical description is presented in Table 2 from which it is clear that the highest rainfall occurs over Momosa and Thyolo of about 1425 mm and 1110 mm respectively. The peak value of 2485 mm occurs over Mwanza. The coefficient of variation varies from 23.4% in Byumbwe to 36.1 in Balaka;
- thus, the variability in the dataset does not change more than 12% from one station to the other. The mean monthly rainfall data is presented in Table 3. From this table, it can be seen that the wet season is from October to March and the highest precipitation occurs over Mimosa during January that is 57.5 mm. Although May, Jun and July are considered dry seasons, they often experience some precipitation, for example, averaging about 10 mm in Mimosa.

#### 4 Result and discussion

## 25 4.2 Annual trend

The results of the trend analysis of annual precipitation using Mann-Kendall and Sen's slope estimate are shown in Table 4. Trend analysis revealed a significant increasing trend of at least 0.001 confidence. The slope of the increase Q ranges from 3.30mm/year at Mwanza to 10.27 mm/year at Mimoza. Trend analysis using linear regression is also presented in Table 5 from which it is clear that the Sen's test results are almost consistent with the linear regression results and reveal the same

conclusion. The spatial distribution of the Z values, shown from Table 4 and Fig. 2, highlight that the highest trend increase was witnessed in Mimosa (Z=3.8) and the lowest in Mwanza (Z=1.69). Time series plots of annual data are presented in

5

Figure 3, from which it is clear that the trend line moves upward at all stations and hence there is an increase in annual rainfall depth. This is also reviled from Table 5, where we can see the slope of the linear regression ranging from 3.4 mm/year at Mwanza to 10.07 mm/year at Mimosa. Additionally, Table 4 highlights the smallest significant level  $\alpha$  with which the test demonstrates that the null hypothesis of no trend should be rejected. For example, in case of Nsanje, the results of the test is returned as H=1, hence the null hypothesis at the significant level of 5% is rejected, whereas in case of Makhanga, the H is zero, thus, the null hypothesis cannot be rejected at the significant level of 5%. The general overview of the table exhibits that the catchment varies in both Z and Q, however, the general observation is that there are significant rising trends at all stations.

#### 4.3 Monthly trend

The MK test was applied to different months and hence every time series for 58 years is considered as uncorrelated and thusmeet the test assumptions. Additionally, since the time lag between any two consecutive data points are virtually one year, then the monthly time series is assumed to be independent.

Table 6 presents the precipitation trend results in terms of the Z statistic for all stations for monthly precipitation. The slope and the intersect of Sen's slope are also presented in Table 7. The test is calculated based on the smallest significant level  $\alpha$  in

- which the trend is considered statistically significant at the  $\alpha$  confidence level. The results determine that most stations exhibit positive trends during wet months (November - March) and negative trends during dry months. Furthermore, it is evident from the graphical illustrations of the Z values in Fig 5. Additionally, the values of z are illustrated using contour lines as seen from figure 6, from which it is clear that the highest values of z are in Monkey Bay during January. The magnitude of the trend is shown from Sen's slope estimate in Table 7. Moreover, it is also clear that the highest magnitude of Sen's slop is in Monkey
- Bay (B=3.67) during January. The numerical results of Z from Table 6, are illustrated using contour lines as in Fig 6, demonstrating that the highest values are around Monkey Bay in the north of the catchment.
- The results of homogeneity of trends between stations based on van Belle and Hughes (1984) at different months are presented in Table 8 and the results of homogeneity of trends between months at different stations are presented in Table 9. The results of the homogeneity of month-station interactions are presented in Table 10. The values of homogeneity Chi-squares for stations, months and station-month interactions were compared with the significant level  $\alpha$ =5%. By comparing the results, it is clear that Chi-squares exceeds the critical value of  $\alpha$  for all cases, thus, the trend results are homogenous between months, stations and between month stations interaction. The homogeneity is also clear from Figure 5, where all stations follow the
- same trend when Z values move from one month to the other. Therefore, it is clear that there is a positive trend during wet seasons and a negative trend during dry seasons.

# 4.4 Peak monthly trend analysis of precipitation

The results of monthly peak trend analysis are presented in Tables 11 and 12. The statistical test Z indicate the there is a positive trend of the peak precipitation during wet months and negative trend during dry months. The highest z value was witnessed in Monkey Bay in the north of about 5.06 during January while the highest negative value is in the same station during Jun of about -5.76. The magnitude of trends is shown from Table 12 from which we can see the Sens' slope ranges from - 0.17 in Chingale to 0.8 in Monkey Bay during the months of January. The general view of the peak monthly tends analysis results show that there is an upward trend on wet seasons in all stations in general and in the northeast part in particular as illustrated in Figure 6. Therefore, the northeast regions may expect more heavy rainfall in the future and hence the area is subject to a more flood.

## 10 5 Conclusion

5

This paper studies the trends in annual, monthly-total and monthly peaks of 16 rainfall stations in the Shire Basin, Malawi, during a period of 58 years, 1953-2010. The techniques applied were based on a nonparametric Mann-Kendall test to detect a monotonic trend of a time series. The magnitude of the trends were estimated with the nonparametric Sen's method. A linear regression model was used to validate the MK significance test and to determine the magnitude of the trend. The spatial and temperal homogeneity of trends were examined using Van Balla and Hughes and ISO MK 7 plots.

temporal homogeneity of trends were examined using Van Belle and Hughes and ISO-MK-Z plots, demonstrating the trend results were homogeneous.

The results highlight that there is an upward trend in annual precipitation. This increase is due to a positive significant trend during wet seasons, (November, December, January and February), in contrast to a down trend during dry seasons. The spatial

distribution of trends increase from south to the north and from east to west. The results also reveal an increase in peak values year over year, which may explain an increase in flood events in recent years. Almost similar trends were observed using Sen's method and linear regression method. Thus, coherent and significant increases in rainfall was observed over wet seasons with obvious decreases found over dry seasons. The cause of these changes, requires further investigation to establish a linkage between climate variability and observed trends.

5