# Peer review of "Spatial and temporal Trend Analysis of Long Term rainfall records in data-poor catchments with missing data, a case study of Lower Shire floodplain in Malawi for the Period 1953-2010"

_Hydrology and Earth System Sciences, 2017_

## Referee Comment (RC1) · Anonymous Referee #1 · 1 Dec 2017

The article investigates trends in monthly rainfall in a basin in Malawi. The manuscript is well written and easy to read, and the study uses well documented statistical methods. Based on my review of the manuscript I only have a few minor remarks: 1. What is the motivation for choosing this basin in Malawi? Why do you make this study on this catchment, and why on this scale? Please develop the text about the geographic and climatologic settings further.

2. Figures 1,2,4 and 6: These maps are quite confusing. To improve comprehension, you might want to consider the following changes: a. Figure 1: Add a zoomed out

map which shows where in the world this place is located for reference. Also, consider highlighting the catchment more, so that it stands out from the rest of the map. Add gauging station numbers and table. b. Figure 2,4,6: Remove gauging station numbers from map and the tables, it is confusing with both station numbers and Z statistics in the same map. Maybe just have green dots indicating the gauging locations? Also, consider only showing the basin here (cut out the rest of Malawi), that would increase the scale of the map and improve legibility.

3. Page 6. Rows 21-23: This is a weird statement, consider re-writing it.

4. The stations are sorted differently in the Tables. Having them all sorted A-Z would make comparisons easier.

---

## Referee Comment (RC2) · Anonymous Referee #2 · 2 Dec 2017

**Review report of the manuscript:**
Spatial and temporal Trend Analysis of Long Term rainfall records in data-poor catchments with missing data, a case study of Lower Shire floodplain in Malawi for the Period 1953-2010
By Rabee Rustum, Adebayo J. Adeloye, and Faidess Mwale

The paper investigates the long-term trends in precipitation from 16 stations located in the lower Shire catchment in Malawi over the period 1953-2010. Both annual and monthly trend analyses are considered. Trend presence of trend is checked using the nonparametric Mann-Kendall (MK) test statistic. The trend magnitude is estimated using the Sen's slope method. Finally, the homogeneity of trends is examined using the Van Belle and Hughes method. According to the authors the results indicate that 20 annual precipitation has increased, whereas, monthly precipitation revealed an upward trend in wet seasons (November to April) and a downward trend in dry seasons (May to October). The monthly peak trend analysis has shown upward trend in rainy months at all stations.

**General comment**
The authors perform very simple analyses using standard tests. In general, this is not to be viewed as a serious drawback. Nonetheless, in cases like this, more than in cases of papers that propose original methodologies, it is indispensable, for the paper to be considered for publication, that adopted methods are applied with scientific rigour and that results are properly interpreted. Unfortunately, in the present case the considered statistical tools are applied in a non-convincing way and interpretation of results is not convincing as well. Valid interpretations can be obtained only if the working assumptions the adopted statistical techniques rely on are met. In the paper, the majority of these working assumptions are not formally checked. In addition some of the considered statistical techniques are not correctly implemented.

**Specific comments**

| | |
|---|---|
| At page 3 lines 4-7
The authors write:
<<The daily records from which the monthly and annual records were generated contain numerous gaps and missing values which require a scientific infilling before performing any trend analysis. 5 This infilling exercise has been performed and reported elsewhere (Mwale, Adeloye, and Rustum 2012)>> | The authors should explain whether (depending on the amount of missing values and/or on the adopted infilling procedure) the infilling exercise can affect or not the result of the trend test. |
| At page 3 lines 23-25
The authors write:
<<It is worth to mention that some researchers (e.g. Sang et al., 2014) highlighted the need for pre-whitening of the time series data before conducting any trend analysis test, however, if the sample size is more than 70 then serial correlation does not affect the MK test (Basistha, 25 Arya, and Goel 2009)>> | It is not clear what the authors mean for sample size. It seems to me that the mentioned condition is not satisfied. For example, the trend analysis of annual precipitation time series is performed on the basis of 58 data. |
| At page 4 line 4
The authors write:
<<…the null hypothesis that a sample data is independent and identically distributed…>> | a) Data is a plural noun.
b) Data are numbers. It is not correct say that numbers are stochastically independent and/or identically distributed.
I suggest to write:
….the hypothesis that observed variables are independent and identically distributed…. |
| At page 4 sentence 5-6
The authors write:
<<Thus, if the null hypothesis $H_0$ is accepted at the significant level $\alpha$, then the mean and variance of the S statistics are given by Kendall as it is approximately normally distributed, mean (S) is zero.>> | a) It is not correct to write significant level. It should be significance level (this error is repeated several times in the manuscript).
b) It is not correct to write:
<<if the null hypothesis $H_0$ is accepted at the significant level α then the mean and variance of the S statistics are given by Kendall >>.
The level of significance only impact on the decision rule. At this stage the authors are not discussing about the test result.
It is correct to say:
Under the null hypothesis $H_0$, the mean and the variance are those given in equations (3) and (4), respectively. |

| | |
|---|---|
| | c) The fact that the mean of $S$ is zero does not depend on the fact that it is approximately normally distributed. It is due to the fact that, under the null hypothesis, the observed variables are assumed independent and identically distributed. In addition, a normally distributed random variable can have a mean that is not zero. |
| At page 4 Equation (5)
The authors write:

$$Var(S) = \left\{ \frac{n(n-1)(2n+5)}{18} - \sum t_i(t_i-1)(2t_i+5) + \right.$$
$$- \sum u_t(u_i-1)(2u_i+5) \right\} +$$
$$+ \frac{1}{9n(n-1)(n-2)}\left\{\sum t_i(t_i-1)(t_i-2)\right\}\left\{\sum u_i(u_i-1)(u_i-2)\right\} +$$
$$+ \frac{1}{2n(n-1)}\left\{\sum t_i(t_i-1)\right\}\left\{\sum u_i(u_i-1)\right\}$$ | a) The symbol $u_i$ is not defined in the text.
b) It is not specified which values can assume the index $i$.
c) This is not the formula usually adopted in the considered case. The authors don't provide a reference for it.
I suggest to write:
$$Var(S) = \frac{1}{18}\left\{n(n-1)(2n+5) - \sum_{i=1}^{m} n_i(n_i-1)(2n_i+5)\right\}$$
where $m$ ($m \geq 2$) indicates the number of tied groups and $n_i$ is the number of elements (i.e., ties) in the $i-th$ group. |
| At page 5 line 1-2
The authors write:
<<However, the absolute value of Z is compared with the standard normal cumulative distribution to detect if there is any trend The trend is said to be decreasing if Z is negative and increasing if Z is positive. H0, the null hypothesis of no trend, is rejected if the absolute value of z is greater than $z_{1-\alpha/2}$, where $z_{1-\alpha/2}$ is obtained from the standard normal cumulative distribution tables.>> | The sentence is misleading. In particular, it is not clear what the authors mean when they write: <<the absolute value of Z is compared with the standard normal cumulative distribution>>.
I suggest to write:
The null hypothesis of no trend, $H_0$ is rejected, at the selected level of significance ($\alpha$), if the absolute value of Z is greater than $z_{1-\alpha/2}$, where $z_{1-\alpha/2}$ is obtained from the standard normal cumulative distribution table. In particular, the trend is said to be decreasing if Z is negative and increasing if Z is positive. |
| **Page 5 section 2.2.**
Sen's slope estimator | As remarked by the authors themselves: <<the method uses a linear model to calculate the change of slope and the variance of the residuals should be constant in time.>>
Although the plots in figure 3 seems to corroborate the validity of these assumptions, the authors should check them via formal tests (and/or at least discuss them explicitly) when they apply the Sen's slope estimator to the considered time series. |
| At page 7 lines 26-28
The authors write:
<<The results of the trend analysis of annual precipitation using Mann-Kendall and Sen's slope estimate are shown in Table 4. Trend analysis revealed a significant increasing trend of at least 0.001 confidence. The slope of the increase Q ranges from 3.30mm/year at Mwanza to 10.27 mm/year at Mimoza.>>
At page 8 lines lines 4-6
The authors write:
<<..Additionally, Table 4 highlights the smallest significant level $\alpha$ with which the test demonstrates that the null hypothesis of no trend should be rejected. For example, in case of Nsanje, the results of the test is returned as H=1, hence the null hypothesis at the significant level of 5% is rejected, whereas in 5 case of Makhanga, the H is zero, thus, the null hypothesis cannot be rejected at the significant level of 5%>>. | a) At page 7 it is not clear what the authors mean for 0.001 confidence. I guess they mean "at the significance level of 0.001". Assumed I am right, this value differ from the significance level of 5% which the authors make reference at page 8.
b) Results in table 4 are not correct. Based on the p-values reported in the table (I checked the p-values, they are correct) only in the case of Mwanza the test is not significant at the significance level of 0.05.
It is clear that something doesn't work. For example, note that, despite of the fact that the values reported in the second row are identical to those reported in third row, according to the authors the null hypothesis is rejected in the case of Makhanga while it is not rejected in the case of Ngabu. |

| | |
|---|---|
| At page 8 lines 9
The authors write:
<<The MK test was applied to different months and hence every time series for 58 years is considered as uncorrelated and thus meet the test assumptions.>> | The meaning of this sentence is not clear. I suspect that the author are trying to say that different time series are not cross-correlated. In the case I am right, I think that they should check this assumption (it is not obvious).
The authors should also explain why this assumption is important for the MK test reported in table 6. I expect it is important for testing the homogeneity of trends (where, for example, it is needed that $Z_j$ computed over different seasons are nearly independent, an assumption that is usually assumed to be reasonable if the seasonal observations are far enough apart). |
| At page 8 lines 14-15
The authors write:
The test is calculated based on the smallest significant level $\alpha$ in which the trend is considered statistically significant at the $\alpha$ confidence level. | a) This sentence is not clear. I suspect the authors are trying to make reference to the p-value.
b) The term confidence level should not be used in the case of tests of hypotheses. It is correct to use the term level of significance. |
| Page 8 lines 24-30
The authors write:
<<The results of homogeneity of trends between stations based on van Belle and Hughes (1984) at different months are presented in Table 8 and the results of homogeneity of trends between months at different stations are presented in Table 9. The results of the homogeneity of month-station interactions are presented in Table 10. The values of homogeneity Chi-squares for stations, months and station-month interactions were compared with the significant level $\alpha$ =5%. By comparing the results, it is clear that Chi-squares exceeds the critical value of $\alpha$ for all cases, thus, the trend results are homogenous between months, stations and between month stations interaction. The homogeneity is also clear from Figure 5, where all stations follow the 30 same trend when Z values move from one month to the other.>> | The null hypotheses of the performed test is that there is homogeneity of trends. In table 9 only the test statistics computed in the case of Mimosa doesn't exceed the critical value (the test statistics has 11 degrees of freedom the critical value is 19.67). In all the remaining cases the right decision, at the significance level of 0.05, is that there is not homogeneity of trends. |
| Page 8 lines 24-30
The authors write:
<<The results of homogeneity of trends between stations based on van Belle and Hughes (1984) at different months are presented in Table 8 and the results of homogeneity of trends between months at different stations are presented in Table 9. The results of the homogeneity of month-station interactions are presented in Table 10. The values of homogeneity Chi-squares for stations, months and station-month interactions were compared with the significant level $\alpha$ =5%. By comparing the results, it is clear that Chi-squares exceeds the critical value of $\alpha$ for all cases, thus, the trend results are homogenous between months, stations and between month stations interaction. The homogeneity is also clear from Figure 5, where all stations follow the 30 same trend when Z values move from one month to the other.>> | Technical language should be revised:
it is not correct to say:
• The values of homogeneity Chi-squares for stations, months and station-month interactions were compared with the significant level $\alpha$ =5%.
It is correct to say significance level. In addition, it is correct to say that the test statistic is compared to the critical value corresponding to the significance level $\alpha$ =5%.
Similarly it is not correct to say:
• The Chi-squares exceeds the critical value of $\alpha$
It is eventually correct to compare the p-value to the significance level. |
| Page 8 lines 24-30
The authors write:
<<The results of homogeneity of trends between stations based on van Belle and Hughes (1984) at different months are presented in Table 8 and the results of homogeneity of trends between months at different stations are presented in Table 9. The results of the homogeneity of month-station interactions are presented in Table 10. The values of homogeneity Chi-squares for stations, months and station-month interactions were compared with the significant level $\alpha$ =5%. By comparing the results, it is clear that Chi-squares exceeds the critical value of $\alpha$ for all cases, thus, the trend results are | It is necessary to specify the number of degrees of freedom of the test statistics considered in tables. For example in table 9 the number of degrees of freedom is 11. |

| | |
|---|---|
| homogenous between months, stations and between month stations interaction. The homogeneity is also clear from Figure 5, where all stations follow the 30 same trend when Z values move from one month to the other.>> | |
| Table 6 | It is not clear the meaning of the asterisks. I guess the number of asterisks increases if the p-value decreases. The authors should clarify this point. |

---

## Author Comment (AC1) · 3 Dec 2017

Thank you very much for the careful review and valuable comments to the initial submission. We will address the comments raised on the initial submission and we will revise the manuscript accordingly to accommodate these comments.

---

## Author Comment (AC2) · 3 Dec 2017

Thank you very much for the careful review and valuable comments to the initial submission. We will address the comments raised on the initial submission and we will revise the manuscript accordingly to accommodate these comments.

[Figure]

Creative Commons CC BY license logo

---

## Referee Comment (RC3) · Anonymous Referee #3 · 15 Jan 2018

This paper shows a trend analysis of some rainfall time series in Malawi. As detailed below, it is my opinion that: 1) The assumptions made and the analyses performed by the Authors are questionable: many of the points made by the Authors have already been thoroughly discussed and questioned in the literature. 2) The paper seems indeed to suffer from a poor literature review: the Authors should demonstrate to the HESS readership how their research fits into the larger field of study. 3) The Authors make strong claims that may well be incorrect and theoretically ungrounded. Consequently, my recommendation to the Editor is to reject this paper. If the Authors wish

to resubmit, I think the amount of required changes goes well beyond a major revision, hence my recommendation of rejection. In the following, I give further details on my major concerns. a) Data gaps are ubiquitous in hydrological time series, and filling these values remains a challenge. The Authors should investigate the influence of their gap filling method on their analysis and provide cautionary remarks for maximal gap allowance. This is especially so for the paper case study, where "several records are missing" (see paper p. 7). b) Hydrological data are commonly characterized by temporal dependence. All trend tests involving the iid hypothesis (such as the Mann-Kendall test used in the paper) should be corrected for the effect of autocorrelation. Neglecting this aspect usually leads to contradictory results further discussed by Khaliq et al. (2009), Bayazit (2015), Serinaldi and Kilsby (2016), Serinaldi et al. (2018) and Tyralis et al. (2018). c) The Authors use the Mann-Kendall test to detect monotonic trends in the observational data, and then quantify their magnitude by the so-called Sen's method, which assumes a linear trend. However, this is not justified. The Mann-Kendall test (MK) refers to monotonic changes that can be either linear or nonlinear. Reducing the indication of possible monotonic trends given by MK to that of a linear trend is too restrictive and somewhat arbitrary, and does not reflect the rationale and outcome of MK test (Serinaldi et al., 2018). d) Based on the Authors' claims (see the paper introduction), a trend of true interest (which also is the focus of the largest part of hydro-meteorological literature on the topic) is related to a form of nonstationarity. However, handling (or even detecting) nonstationarity merely from data may be difficult, if not impossible. I endorse herein the following statement by Koutsoyiannis and Montanari (2015): "To establish a deterministic function of time, as required in order to claim nonstationarity, we need at least both of the following conditions to hold: (a) deductive reasoning in order to establish the deterministic function of time; (b) validation of the deterministic function by data which were not used in the model construction". Hence, testing trends on observed time series can easily be inconclusive and/or misleading because of the intrinsic difficulty, if not impossibility, of detecting nonstationarity (of a process) solely from data without exogenous information (Serinaldi et al., 2018). e)

Based on the considerations above, the following statement in the paper conclusions (p. 9) is misleading and theoretically ungrounded: "Thus, coherent and significant increases in rainfall was observed over wet seasons with obvious decreases found over dry seasons. The cause of these changes, requires further investigation to establish a linkage between climate variability and observed trends". As a trend is a systematic change reflecting a time-dependent process, the mathematical rule describing the evolution of this change should be established a priori (see e.g., Poppick et al., 2017).

REFERENCES Bayazit, M. (2015). Nonstationarity of hydrological records and recent trends in trend analysis: a state-of-the-art review. Environmental Processes, 2(3), 527-542.

Khaliq, M. N., Ouarda, T. B., Gachon, P., Sushama, L., & St-Hilaire, A. (2009). Identification of hydrological trends in the presence of serial and cross correlations: A review of selected methods and their application to annual flow regimes of Canadian rivers. Journal of Hydrology, 368(1-4), 117-130.

Koutsoyiannis, D., & Montanari, A. (2015). Negligent killing of scientific concepts: the stationarity case. Hydrological Sciences Journal, 60 (7-8), 1174–1183.

Poppick, A., Moyer, E. J., & Stein, M. L. (2017). Estimating trends in the global mean temperature record. Advances in Statistical Climatology, Meteorology and Oceanography, 3, 33-53.

Serinaldi, F., & Kilsby, C. G. (2016). The importance of prewhitening in change point analysis under persistence. Stochastic Environmental Research and Risk Assessment, 30(2), 763-777.

Serinaldi, F., Kilsby, C. G., & Lombardo, F. (2018). Untenable nonstationarity: An assessment of the fitness for purpose of trend tests in hydrology. Advances in Water Resources, 111, 132-155.

Tyralis, H., Dimitriadis, P., Koutsoyiannis, D., O'Connell, P. E., Tzouka, K., & Iliopoulou,

T. (2018). On the long-range dependence properties of annual precipitation using a global network of instrumental measurements. Advances in Water Resources, 111, 301-318.